# First-Trimester Screening for Miscarriage or Stillbirth—Prediction Model Based on MicroRNA Biomarkers

**DOI:** 10.3390/ijms241210137

**Published:** 2023-06-14

**Authors:** Ilona Hromadnikova, Katerina Kotlabova, Ladislav Krofta

**Affiliations:** 1Department of Molecular Biology and Cell Pathology, Third Faculty of Medicine, Charles University, 14700 Prague, Czech Republic; katerina.kotlabova@lf3.cuni.cz; 2Institute for the Care of the Mother and Child, Third Faculty of Medicine, Charles University, 14700 Prague, Czech Republic; ladislav.krofta@upmd.eu

**Keywords:** cardiovascular diseases, first-trimester screening, gene expression, microRNA, miscarriage, prediction, stillbirth, whole peripheral venous blood

## Abstract

We evaluated the potential of cardiovascular-disease-associated microRNAs to predict in the early stages of gestation (from 10 to 13 gestational weeks) the occurrence of a miscarriage or stillbirth. The gene expressions of 29 microRNAs were studied retrospectively in peripheral venous blood samples derived from singleton Caucasian pregnancies diagnosed with miscarriage (*n* = 77 cases; early onset, *n* = 43 cases; late onset, *n* = 34 cases) or stillbirth (*n* = 24 cases; early onset, *n* = 13 cases; late onset, *n* = 8 cases; term onset, *n* = 3 cases) and 80 selected gestational-age-matched controls (normal term pregnancies) using real-time RT-PCR. Altered expressions of nine microRNAs (upregulation of miR-1-3p, miR-16-5p, miR-17-5p, miR-26a-5p, miR-146a-5p, and miR-181a-5p and downregulation of miR-130b-3p, miR-342-3p, and miR-574-3p) were observed in pregnancies with the occurrence of a miscarriage or stillbirth. The screening based on the combination of these nine microRNA biomarkers revealed 99.01% cases at a 10.0% false positive rate (FPR). The predictive model for miscarriage only was based on the altered gene expressions of eight microRNA biomarkers (upregulation of miR-1-3p, miR-16-5p, miR-17-5p, miR-26a-5p, miR-146a-5p, and miR-181a-5p and downregulation of miR-130b-3p and miR-195-5p). It was able to identify 80.52% cases at a 10.0% FPR. Highly efficient early identification of later occurrences of stillbirth was achieved via the combination of eleven microRNA biomarkers (upregulation of miR-1-3p, miR-16-5p, miR-17-5p, miR-20a-5p, miR-146a-5p, and miR-181a-5p and downregulation of miR-130b-3p, miR-145-5p, miR-210-3p, miR-342-3p, and miR-574-3p) or, alternatively, by the combination of just two upregulated microRNA biomarkers (miR-1-3p and miR-181a-5p). The predictive power achieved 95.83% cases at a 10.0% FPR and, alternatively, 91.67% cases at a 10.0% FPR. The models based on the combination of selected cardiovascular-disease-associated microRNAs had very high predictive potential for miscarriages or stillbirths and may be implemented in routine first-trimester screening programs.

## 1. Introduction

Miscarriage (spontaneous abortion) is defined as pregnancy loss before 20 weeks of gestation [1]. Miscarriage occurs approximately in 15.0–30.0% of all pregnancies [2,3,4] and in 8–15% of pregnancies confirmed by ultrasonography and histopathology [5,6]. The loss of a pregnancy before 13 completed weeks (12 6/7) defined as an empty gestational sac or a lack of fetal heartbeat is usually called early pregnancy loss [7].

A late miscarriage, also called a second-trimester or mid-trimester pregnancy loss, is one that happens within the range between 13 and 20 weeks of gestation [8,9,10,11,12].

Recurrent pregnancy loss (RPL) is defined as two or more consecutive losses of pregnancy occurring before 20 weeks of gestation and usually affects 2–5% of pregnancies [13,14] if nonvisualized and unconfirmed pregnancies are also included. If only pregnancy losses confirmed by ultrasonography and histopathology are counted, the prevalence of RPL is considerably lower, ranging from 0.8 to 1.4% [11,15,16].

Stillbirth is defined as an intrauterine fetal death after 20 gestational weeks [17,18], when a fetus has no signs of life due to the absence of breathing, heartbeat, pulsation of the umbilical cord, or definite movements of voluntary muscles [18,19]. If the gestational age is unknown, a weight of 350 g or more is usually used as a criterion [18].

Early stillbirth occurs between 20 and 27 completed weeks of pregnancy, late stillbirth occurs between 28 and 36 completed pregnancy weeks, and term stillbirth occurs after 37 or more completed pregnancy weeks.

The incidence of stillbirths differs between developed countries (2.0–5.3 per thousand deliveries) and developing countries (25.5–40.0 per thousand deliveries) [20,21].

A whole range of studies have evaluated the potential of current or novel first-trimester biochemical and metabolomic biomarkers or hemostatic parameters to identify pregnancies at risk of spontaneous abortion [22,23,24] or stillbirth [25]. In addition, early first-trimester ultrasound characteristics, such as low fetal heart rate and small crown–rump length, were identified to be associated with a higher risk of subsequent pregnancy loss [3].

Plenty of models have been developed to predict miscarriages or stillbirths [26,27,28,29,30,31]. These models are usually based on the combination of maternal characteristics and history with ultrasound variables and biochemical biomarkers. Nevertheless, since most of these models have not yet been internally or externally validated and clinically evaluated, they are not usable in routine practice [26,32].

A previously developed simple scoring system to predict miscarriage in intrauterine pregnancies of uncertain viability [33] was finally successfully externally validated [34]. This system was based on the combination of basic clinical and ultrasound features, such as maternal age, bleeding score, mean gestational sac diameter, and the presence or absence of a yolk sac [33,34]. A simple scoring system to predict early pregnancy viability beyond the first trimester was also independently externally validated in a population of Chinese pregnant women [35].

A recently developed and validated model using a combination of maternal factors, estimated fetal weight, and the uterine artery pulsatility index was able to predict at 22 gestational weeks the occurrence of placental-dysfunction-related stillbirth before 37 gestational weeks in 62.3% of cases at a 10.0% false positive rate (FPR). However, it was not able to predict unexplained stillbirths or stillbirths occurring due to other reasons [36].

Stillbirths usually result from various etiologies. They are multifactorial, and therefore, it is almost impossible to predict at early stages of gestation all of the types and causes of stillbirths [27].

Recently, we observed altered expression profiles of microRNAs playing a role in the development and the maintenance of homeostasis of the cardiovascular system and in the pathophysiology of cardiovascular and cerebrovascular diseases in pregnancies at risk of adverse pregnancy outcome. Based on these findings, we proposed efficient early predictive models for gestational hypertension (GH) [37], preeclampsia (PE) [37], HELLP syndrome [38], fetal growth restriction (FGR) [39], small gestational age (SGA) [39], preterm delivery [40], and/or gestational diabetes mellitus (GDM) [41].

Therefore, we are interested to see if altered expression profiles of cardiovascular-disease-associated microRNAs (increased or decreased levels of microRNAs) are also present in pregnancies at risk of miscarriage or stillbirth.

MicroRNAs belong to the family of small noncoding RNAs (18–25 nucleotides) that regulate gene expression at the posttranscriptional level by degrading or blocking the translation of target messenger RNA (mRNA). Increased microRNA expression suggests the downregulation of potential target genes, and vice versa, decreased microRNA expression is usually associated with the upregulation of potential target genes, which may contribute to the pathophysiology of a disease.

We are interested to explore if it is feasible to develop an efficient early predictive model for miscarriage or stillbirth based on altered expressions of microRNAs playing key roles in the functioning of the cardiovascular system [38].

## 2. Results

### 2.1. Identification of Risk Factors Predisposed to Miscarriage/Stillbirth

The clinical characteristics of the cases (pregnancies with miscarriages or stillbirths in current gestation) and controls (pregnancies with normal courses of gestation) are indicated in Table 1.

Multiple independent risk factors predisposed to miscarriages or stillbirths were identified in our studied group: advanced maternal age, obesity, nonautoimmune hypothyroidism, confirmed diagnosis of autoimmune disease, the presence of thrombophilic gene mutations, infertility treatment by assisted reproductive technology, nulliparity, smoking of cigarettes during gestation or in the past, the presence of uterine fibroids or an abnormal-shaped womb, and cervical incompetence (a weakened cervix).

Positive first-trimester screening for PE and/or FGR, as well as for spontaneous preterm delivery occurring before 34 gestational weeks, as assessed by Fetal Medicine Foundation (FMF) algorithms [42] also represented independent risk factors for miscarriages or stillbirths.

Perinatal death in anamnesis showed only a trend toward statistical significance in our study.

The statistical data showed that pregnancies with a later occurrence of stillbirth had undergone more frequent invasive prenatal diagnosis procedures (chorionic villus sampling or amniocentesis) on the basis of positive results for the first or the second prenatal screening tests.

No renal diseases, epilepsy, severe anemia, hyperthyroidism, or cholestasis of pregnancy were present in our studied group. No pregnancy suffered from sexually transmitted infections.

### 2.2. Combination of MicroRNA Biomarkers Differentiating between Miscarriage/Stillbirth and Normal Term Pregnancies in Early Stages of Gestation

Gene expression profiles of microRNAs were compared in the early stages of gestation between pregnancies with subsequent miscarriages or stillbirths and normal term pregnancies.

While the levels of miR-1-3p (*p* < 0.001 ***), miR-16-5p (*p* < 0.001 ***), miR-17-5p (*p* < 0.001 ***), miR-26a-5p (*p* = 0.007 *), miR-146a-5p (*p* < 0.001 ***), and miR-181a-5p (*p* < 0.001 ***) were significantly increased, the levels of miR-130b-3p (*p* < 0.001 ***), miR-342-3p (*p* < 0.001 ***), and miR-574-3p (*p* = 0.001 **) were significantly decreased in pregnancies with miscarriages or stillbirths (Figure 1).

MiR-1-3p (54.46%), miR-16-5p (43.56%), and miR-181a-5p (46.53%) differentiated between cases and controls with very good sensitivity at a 10.0% FPR. MiR-17-5p (30.69%), miR-26a-5p (17.82%), miR-130b-3p (23.76%), miR-146a-5p (31.68%), miR-342-3p (26.73%), and miR-574-3p (15.84%) demonstrated a bit lower sensitivity at a 10.0% FPR (Appendix A).

The combination of nine microRNA biomarkers with altered expressions in early stages of gestation (miR-1-3p, miR-16-5p, miR-17-5p, miR-26a-5p, miR-130b-3p, miR-146a-5p, miR-181a-5p, miR-342-3p, and miR-574-3p) identified pregnancies with subsequent miscarriages or stillbirths with an excellent accuracy. A total of 99.01% of cases were revealed at a 10.0% FPR (AUC 0.992, *p* < 0.001, 99.01% sensitivity, 93.75% specificity, cut-off > 0.3262) (Figure 1).

### 2.3. Combination of MicroRNA Biomarkers Differentiating between Miscarriages and Normal Term Pregnancies in Early Stages of Gestation

Next, increased levels of miR-1-3p (*p* < 0.001 ***), miR-16-5p (*p* < 0.001 ***), miR-17-5p (*p* < 0.001 ***), miR-26a-5p (*p* = 0.006 *), miR-146a-5p (*p* < 0.001 ***), and miR-181a-5p (*p* < 0.001 ***), as well as decreased levels of miR-130b-3p (*p* < 0.001 ***) and miR-195-5p (*p* = 0.002 **) were observed during early stages of gestation in pregnancies with subsequent miscarriages (Appendix A).

The combination of these eight microRNA biomarkers with altered expressions in early stages of gestation was able to correctly identify 80.52% of pregnancies, regardless of the onset of miscarriage (early and late miscarriages), at a 10.0% FPR (AUC 0.950, *p* < 0.001, 96.10% sensitivity, 80.00% specificity, cut-off > 0.3668) (Figure 2).

### 2.4. Combination of MicroRNA Biomarkers Differentiating between Early Miscarriages, Late Miscarriages, and Normal Term Pregnancies in Early Stages of Gestation

Increased levels of miR-1-3p (*p* < 0.001 ***, *p* < 0.001 ***), miR-16-5p (*p* < 0.001 ***, *p* < 0.001 ***), miR-17-5p (*p* < 0.001 ***, *p* < 0.001 ***), miR-146a-5p (*p* = 0.001 **, *p* = 0.022 *), and miR-181a-5p (*p* < 0.001 ***, *p* < 0.001 ***), as well as decreased levels of miR-130b-3p (*p* < 0.001 ***, *p* < 0.001 ***), were observed during early stages of gestation in pregnancies, with early miscarriages occurring before 13 gestational weeks and late miscarriages occurring between 13 and 20 gestational weeks. In addition, decreased levels of miR-195-5p (*p* = 0.004 *) were found in pregnancies with early miscarriages only (Appendix A).

The combination of seven microRNA biomarkers with altered expressions in early stages of gestation was able to correctly identify 86.05% of pregnancies with an early miscarriage at a 10.0% FPR (AUC 0.959, *p* < 0.001, 93.02% sensitivity, 85.00% specificity, cut-off > 0.2801) (Figure 3).

The combination of six microRNA biomarkers with altered expressions in early stages of gestation was able to correctly identify 79.41% of pregnancies with a late miscarriage at a 10.0% FPR (AUC 0.941, *p* < 0.001, 94.12% sensitivity, 77.50% specificity, cut-off > 0.2017) (Figure 4).

### 2.5. Combination of MicroRNA Biomarkers Differentiating between Stillbirths and Normal Term Pregnancies

In addition to increased levels of miR-1-3p (*p* < 0.001 ***), miR-16-5p (*p* < 0.001 ***), miR-17-5p (*p* < 0.001 ***), miR-20a-5p (*p* < 0.001 ***), miR-146a-5p (*p* < 0.001 ***), and miR-181a-5p (*p* < 0.001 ***), decreased levels of miR-130b-3p (*p* < 0.001 ***), miR-145-5p (*p* = 0.008 *), miR-210-3p (*p* < 0.001 ***), miR-342-3p (*p* < 0.001 ***), and miR-574-3p (*p* < 0.001 ***) were found during early stages of gestation in pregnancies with stillbirths (Appendix A).

The combination of 11 microRNA biomarkers with altered expressions in early stages of gestation was able to correctly identify 95.83% of pregnancies with stillbirths at a 10.0% FPR (AUC 0.986, *p* < 0.001, 95.83% sensitivity, 98.75% specificity, cut-off > 0.2839) (Figure 5).

Effective screening for stillbirth was also possible using the combination of two microRNA biomarkers only. The combination of miR-1-3p and miR-181a-5p revealed at early stages of gestation 91.67% of cases at a 10.0% FPR (AUC 0.951, *p* < 0.001, 91.67% sensitivity, 93.75% specificity, cut-off > 0.2382) (Figure 6).

### 2.6. Correlation between microRNA Gene Expression and Maternal Age and BMI

In this set of patients, we observed weak positive correlations between microRNA expression and maternal age (miR-1-3p, ρ = 0.219, *p* = 0.003; miR-16-5p, ρ = 0.307, *p* < 0.001; miR-17-5p, ρ = 0.207 *p* = 0.005; miR-26a-5p, ρ = 0.252, *p* < 0.001; miR-145-5p, ρ = 0.161, *p* = 0.030; miR-146a-5p, ρ = 0.264, *p* < 0.001; miR-181a-5p, ρ = 0.238, *p* = 0.001; and miR-574-3p, ρ = 0.178, *p* = 0.016).

In addition, in this set of patients we observed a weak positive correlation between the expression of miR-20a-5p and BMI (ρ = 0.173, *p* = 0.023).

## 3. Discussion

Firstly, we were interested in the identification of risk factors associated with miscarriages and stillbirths in our group of pregnant women.

As with many other researchers, we identified the associations between multiple risk factors, such as an advanced maternal age [18,30,43,44], obesity [18,28,30,45,46,47], smoking of cigarettes [18,30,48], nulliparity [18,49], current pregnancy arisen using assisted reproductive technologies [18,30,50,51,52,53,54], past obstetric history involving a previous stillbirth [18,30,55], the presence of autoimmune diseases [18], the presence of uterine fibroids or an abnormal-shaped womb [56,57,58,59], the presence of cervical incompetence (a weakened cervix) [60], and the occurrence of a miscarriage or stillbirth.

Only uncontrolled thyroid disease was previously suspected to be associated with stillbirth [18]. Nevertheless, it has been proved that pregnant women with thyroid dysfunction or thyroid antibodies also have an increased risk of miscarriage or stillbirth [61,62,63,64,65]. The results of our study may support these findings since we also observed an association between the occurrence of nonautoimmune hypothyroidism, even if it was well-controlled, and miscarriage or stillbirth.

Currently, the American College of Obstetricians and Gynecologists (ACOG) recommends the performance of screening for thrombophilia in patients with more than three miscarriages, late miscarriages, or fetal deaths [66]. There is no doubt that thrombophilia can lead to pregnancy complications, including miscarriage, FGR, and stillbirth [67,68]. In agreement with these findings, we also observed a significantly higher incidence of thrombophilic gene mutations, including factor V Leiden mutation and factor II mutation, in pregnancies with miscarriage or stillbirth.

In addition, in our case-control study we also detected a higher incidence of miscarriage or stillbirth within pregnancies with positivity of first-trimester screening for PE and/or FGR, as well as spontaneous preterm delivery occurring before 34 gestational weeks, all assessed by FMF algorithms [42]. These findings are in compliance with studies of other researchers that have developed and validated models for the prediction of miscarriage and stillbirth during 11–13 gestational weeks [29,30]. These models are based on combinations of parameters similar to the models for the prediction of PE, FGR, and spontaneous preterm delivery occurring before 34 gestational weeks.

Furthermore, we were interested if it was feasible to develop an efficient early predictive model for miscarriage or stillbirth based on altered expressions of microRNAs playing key roles in the functioning of the cardiovascular system [38]. We were interested to explore if microRNA biomarkers would only be able to differentiate sufficiently between pregnancies with miscarriage or stillbirth and normal term pregnancies or if it would be necessary to use a combination of microRNA biomarkers with other risk factors identified in our group of pregnancies with miscarriages and stillbirths to achieve satisfactory discrimination power.

Surprisingly, the combination of nine microRNA biomarkers showed a relatively high accuracy for the early identification of pregnancies with miscarriages or stillbirths. A total of 99.01% of the cases were revealed at a 10.0% FPR. Similarly, a combination of microRNA biomarkers was able to correctly identify 80.52% of pregnancies with miscarriages (eight microRNA biomarkers used) and 95.83% of pregnancies with stillbirths (eleven microRNA biomarkers used) at a 10.0% FPR. Alternatively, if only two microRNA biomarkers were used, 91.67% of pregnancies with stillbirths were identified at a 10.0% FPR. In parallel, pregnancies with early miscarriages (86.05% of cases at a 10.0% FPR) or late miscarriages (79.41% of cases at a 10.0% FPR) were identified using a combination of microRNA biomarkers (seven microRNA biomarkers used in the case of early miscarriage, and six microRNA biomarkers in the case of late miscarriage).

Since a very high discrimination power was achieved to identify early pregnancies at risk of miscarriage or stillbirth with the usage of microRNA biomarkers only, we came to the conclusion that there was no need for the further implementation of additional risk factors into the predictive algorithms, such as maternal clinical characteristics.

To the best of our knowledge, multiple studies on the early identification of miscarriages [69,70,71] or recurrent spontaneous abortions [72,73,74,75,76,77,78,79,80] via the screening of circulating microRNAs in maternal plasma/serum or peripheral blood samples are currently available. Unfortunately, the expression profiles of the majority of microRNAs, which are the subject of our interest, have not previously been studied in patients with miscarriages or recurrent spontaneous abortions. In addition, a limited number of studies on microRNAs (miR-23a-3p, miR-29a-3p, miR-100-5p, and miR-221-3p) whose gene expressions have been examined in patients with miscarriages or recurrent spontaneous abortions are currently available.

MiR-221-3p has been found either not to be dysregulated [72] or upregulated in plasma/serum samples of pregnancies with recurrent spontaneous abortions [76,80]. Our study showed that the expression profile of miR-221-3p was not changed in early stages of gestation in patients with miscarriages or stillbirths.

Several studies have reported decreased plasma levels of miR-23a-3p in pregnancies with idiopathic recurrent pregnancy loss [73] and increased plasma levels of miR-29a-3p and miR-100-5p in pregnancies with recurrent miscarriages [79]. A lower expression of miR-23a-3p and higher expressions of miR-29a-3p and miR-100-5p were not observed in pregnancies with miscarriages or stillbirths in our study. It is possible that only the recurrence of pregnancy loss may lead to the dysregulation of these particular microRNAs.

The study of Ding et al. [81] indicated an aberrant expression of miR-146a-5p in placental villous tissues derived from patients with recurrent spontaneous abortions. They reported a negative correlation between the expression levels of miR-146a-5p (over-expression) and TNF-receptor-associated factor 6 (TRAF6, downregulation) and suggested that this mechanism might play a role in the suppression of trophoblast migration and invasion in patients with recurrent spontaneous abortions. This finding may support our results indicating increased levels of miR-146a-5p in early stages of gestation in pregnancies with miscarriages or stillbirths.

Unfortunately, no studies on the early prediction of stillbirth via the screening of circulating microRNAs in maternal plasma/serum or peripheral blood samples are currently available.

## 4. Materials and Methods

### 4.1. Patient Cohort

Miscarriages (spontaneous abortions) were diagnosed in 77 pregnancies, in which early pregnancy loss occurred in 43 cases and late miscarriages in 34 cases.

The diagnosis of recurrent pregnancy loss (two or more consecutive losses of pregnancy occurring before 20 weeks of gestation) was confirmed in 2 cases before the start of the ongoing pregnancy.

The potential causes of miscarriage could be the following: chromosomal or genetic abnormalities (*n* = 3), chorioamnionitis (*n* = 8), placental insufficiency (*n* = 2), subchorionic/retroplacental hematoma (*n* = 2), or a cervical incompetence (*n* = 2). In 60 cases, the miscarriages remained unexplained.

Stillbirths were diagnosed in 24 pregnancies, in which early stillbirths occurred in 13 cases, late stillbirths in 8 cases, and term stillbirths in 3 cases.

A history of stillbirth, a risk factor for recurrent fetal death in a subsequent pregnancy, was identified in 2 cases.

Pregnancies were categorized as viable (*n* = 95) or miscarriage (*n* = 6) at the time of the first-trimester screening based on ultrasound findings.

The diagnoses of miscarriage (spontaneous abortion) and early and late pregnancy losses were assessed using the ACOG guidelines [7].

The diagnoses of stillbirth as early/late/term stillbirths were assessed using the ACOG guidelines [18].

The potential causes of stillbirth could be the following: chromosomal and genetic abnormalities (*n* = 2), chorioamnionitis (*n* = 6), placental abruption (*n* = 3), severe forms of fetal growth restriction (*n* = 2), or umbilical cord events (*n* = 6). In 5 cases, the stillbirths remained unexplained.

The control group consisted of 80 normal term pregnancies (delivery of a healthy infant after completing 37 weeks of gestation with a weight above 2500 g) selected on the basis of the equality of gestational age at sampling and the storage times of biological samples.

### 4.2. Collection and Processing of Samples

Collection and processing of the samples, reverse transcription (RT), and real-time PCR analyses were performed as previously described [37,38,39,40,41].

Whole peripheral venous blood (EDTA) samples were collected from singleton Caucasian pregnancies between 10 and 13 gestational weeks within the period of 11/2012–8/2021. In total, 8250 samples were collected. Finally, complete medical records were available in 5905 pregnancies, in which 77 pregnancies were diagnosed with miscarriages and 24 pregnancies with stillbirths.

In detail, leukocyte lysates were prepared from 200 µL peripheral blood samples using a QIAamp RNA Blood Mini Kit (Qiagen, Hilden, Germany) and were stored in a mixture of RLT buffer and β-mercaptoethanol (β-ME) at −80 °C.

A MirVana microRNA Isolation kit (Ambion, Austin, TX, USA) was used to isolate the RNA fraction and was highly enriched for small RNAs.

Gene expressions of microRNAs associated with the cardiovascular system (miR-1-3p, miR-16-5p, miR-17-5p, miR-20a-5p, miR-20b-5p, miR-21-5p, miR-23a-3p, miR-24-3p, miR-26a-5p, miR-29a-3p, miR-92a-3p, miR-100-5p, miR-103a-3p, miR-125b-5p, miR-126-3p, miR-130b-3p, miR-133a-3p, miR-143-3p, miR-145-5p, miR-146-5p, miR-155-5p, miR-181a-5p, miR-195-5p, miR-199a-5p, miR-210-3p, miR-221-3p, miR-342-3p, miR-499a-5p, and miR-574-3p) were studied.

RT and real-time qPCR analyses were performed using TaqMan MicroRNA Assays (Applied Biosystems, Branchburg, NJ, USA) with a 7500 Real-Time PCR System under standard TaqMan PCR conditions.

The relative expressions of microRNA genes were assessed using the delta-delta Ct method [82]. Endogenous controls (RNU58A and RNU38B) were used to normalize the microRNA gene expression data [83,84].

### 4.3. Statistical Analysis

MicroRNA gene expressions were compared between cases and controls using the Mann–Whitney test and the Kruskal–Wallis one-way analysis of variance. Afterward, post hoc tests for comparisons among groups and Benjamini–Hochberg correction were applied [85].

Box plots displaying the median, the 75th and 25th percentiles, outliers (circles), and extremes (asterisks) were produced with Statistica software (version 9.0; StatSoft, Inc., Tulsa, OK, USA).

Receiver operating characteristic (ROC) curves displayed the area under the curve (AUC), the cut-off points associated with sensitivities, specificities, positive and negative likelihood ratios (LR+, LR−), and sensitivities at a 10.0% false positive rate (FPR) (MedCalc Software bvba, Ostend, Belgium).

To select the best microRNA combinations, logistic regressions with subsequent ROC curve analyses were applied (MedCalc Software bvba, Ostend, Belgium).

Correlations between variables, including relative microRNA quantification and maternal age and BMI, were calculated using the Spearman’s rank correlation coefficient (ρ). If the correlation coefficient value ranged within <0; 0.5>, there was a weak positive correlation. The significance level was established at a *p*-value of *p* < 0.05.

## 5. Conclusions

Consecutive large-scale retrospective and prospective analyses have to be performed to verify the reliability of cardiovascular-disease-associated microRNA biomarkers to differentiate at early stages of gestation between pregnancies with the subsequent occurrence of miscarriage or stillbirth and pregnancies with a normal course of gestation. In the case that the satisfactory discrimination power is confirmed, gynecologists and obstetricians could have at their disposal at early stages of gestation an easy test to identify pregnancies at risk of miscarriage or stillbirth. This approach may be beneficial in cases with a risk of stillbirth where making a well-timed decision on delivery by induction or Caesarean section may prevent stillbirth. In the case of a risk of miscarriage, pregnancies that may be categorized as risky pregnancies may be sent from early stages of gestation to special prenatal units providing more advanced care. By using this approach, some miscarriages, especially those ones with cervical incompetence, may be avoided.

## 6. Patents

National patent application-Industrial Property Office, Czech Republic (Patent n. PV 2022-505).

## Figures and Tables

**Figure 1 ijms-24-10137-f001:**
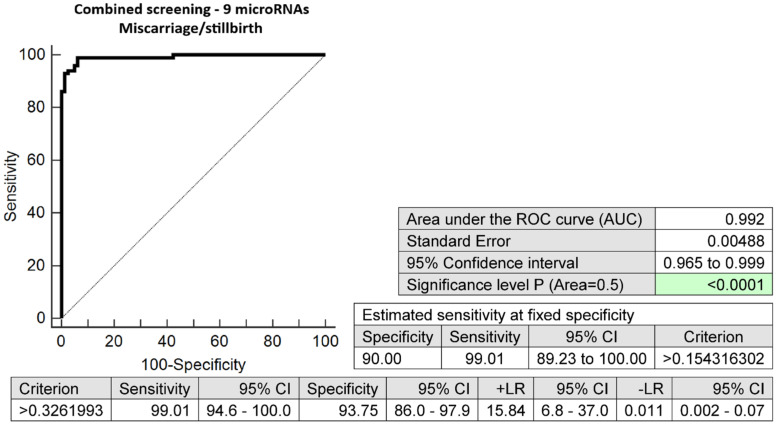
ROC analysis—the combination of 9 microRNA biomarkers (miR-1-3p, miR-16-5p, miR-17-5p, miR-26a-5p, miR-130b-3p, miR-146a-5p, miR-181a-5p, miR-342-3p, and miR-574-3p) was able to identify correctly at early stages of gestation 99.01% of pregnancies with subsequent miscarriages or stillbirths at a 10.0% FPR.

**Figure 2 ijms-24-10137-f002:**
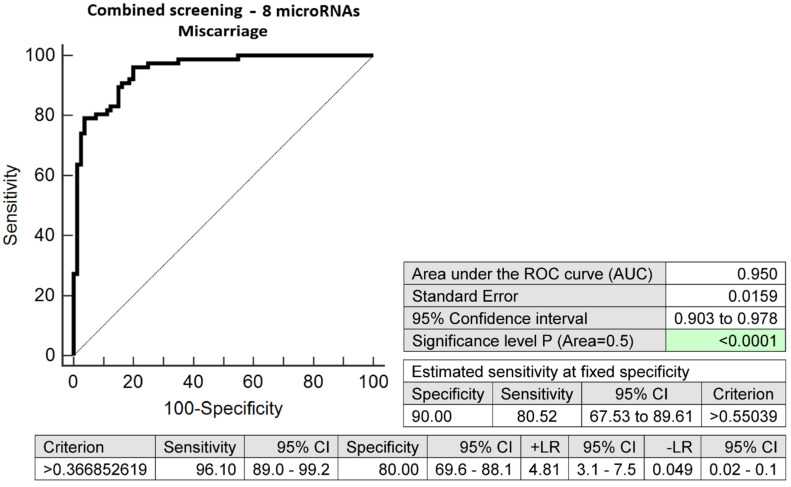
ROC analysis—the combination of 8 microRNA biomarkers (miR-1-3p, miR-16-5p, miR-17-5p, miR-26a-5p, miR-130b-3p, miR-146a-5p, miR-181a-5p, and miR-195-5p) was able to identify correctly at early stages of gestation 80.52% of pregnancies with miscarriages (early and late miscarriages) at a 10.0% FPR.

**Figure 3 ijms-24-10137-f003:**
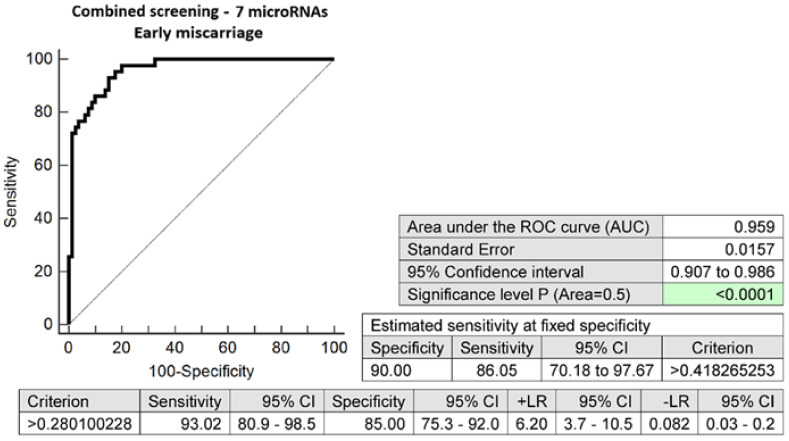
ROC analysis—the combination of 7 microRNA biomarkers (miR-1-3p, miR-16-5p, miR-17-5p, miR-130b-3p, miR-146a-5p, miR-181a-5p, and miR-195-5p) was able to identify correctly at early stages of gestation 86.05% of pregnancies with an early miscarriage at a 10.0% FPR.

**Figure 4 ijms-24-10137-f004:**
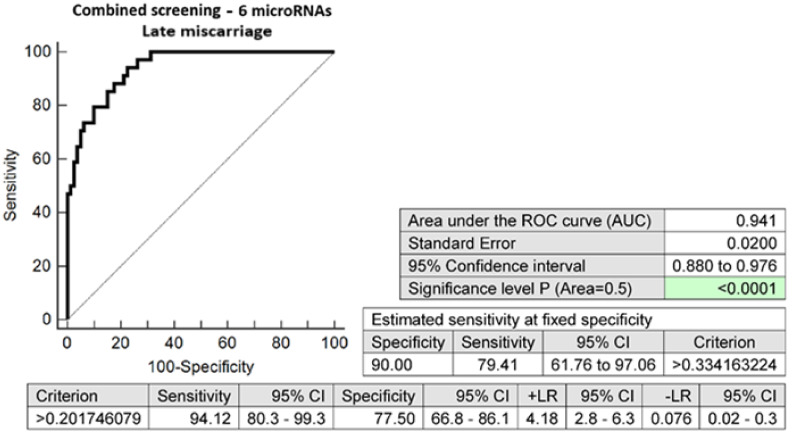
ROC analysis—the combination of 6 microRNA biomarkers (miR-1-3p, miR-16-5p, miR-17-5p, miR-130b-3p, miR-146a-5p, and miR-181a-5p) was able to identify correctly at early stages of gestation 79.41% of pregnancies with a late miscarriage at a 10.0% FPR.

**Figure 5 ijms-24-10137-f005:**
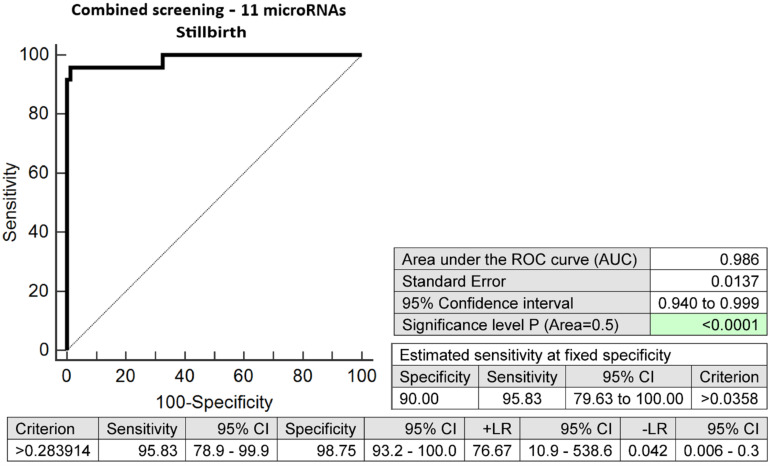
ROC analysis—the combination of 11 microRNA biomarkers (miR-1-3p, miR-16-5p, miR-17-5p, miR-20a-5p, miR-130b-3p, miR-145-5p, miR-146a-5p, miR-181a-5p, miR-210-3p, miR-342-3p, and miR-574-3p) was able to identify correctly at early stages of gestation 95.83% of pregnancies with stillbirths at a 10.0% FPR.

**Figure 6 ijms-24-10137-f006:**
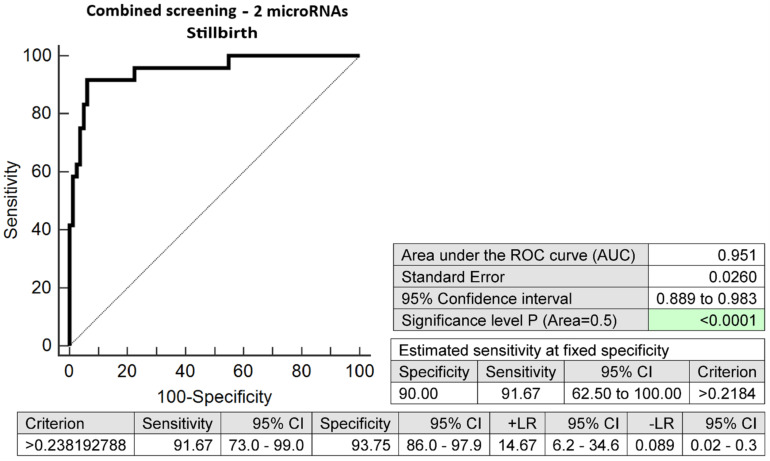
ROC analysis—the combination of 2 microRNA biomarkers (miR-1-3p and miR-181a-5p) was able to identify correctly at early stages of gestation 91.67% of pregnancies with stillbirths at a 10.0% FPR.

**Table 1 ijms-24-10137-t001:** Clinical characteristics of the cases and controls.

	Normal Term Pregnancies (*n* = 80)	Miscarriages or Stillbirths (*n*= 101)	Miscarriages (*n* = 77)	Stillbirths (*n* = 24)	*p*-Value ^1^	*p*-Value ^2^	*p*-Value ^3^
*Maternal history*							
Venous thromboembolism in history	0 (0%)	2 (1.98%)	2 (3.90%)	0 (0%)	0.369 OR: 4.045 95% CI: 0.191–85.468	0.283 OR: 5.331 95% CI: 0.252–112.863	0.555 OR: 3.286 95% CI: 0.063–169.964
Diabetes mellitus (T1DM, T2DM)	0 (0%)	1 (0.99%) 1 T1DM	1 (1.30%) 1 T1DM	0 (0%)	0.593 OR: 2.403 95% CI: 0.096–59.786	0.483 OR: 3.157 95% CI: 0.127–78.689	0.555 OR: 3.286 95% CI: 0.063–169.964
Chronic hypertension on treatment	0 (0%)	2 (1.98%)	1 (1.30%)	1 (4.17%)	0.369 OR: 4.045 95% CI: 0.191–85.468	0.483 OR: 3.157 95% CI: 0.127–78.689	0.158 OR: 10.277 95% CI: 0.405–260.697
Congenital heart disease	0 (0%)	1 (0.99%) 1 WPW syndrome	0 (0%)	1 (4.17%) 1 WPW syndrome	0.593 OR: 2.403 95% CI: 0.096–59.786	0.985 OR: 1.039 95% CI: 0.020–53.004	0.158 OR: 10.277 95% CI: 0.405–260.697
Nonautoimmune hypothyroidism	2 (2.50%)	11 (10.89%)	9 (11.69%)	2 (8.33%)	0.046 OR: 4.767 95% CI: 1.025–22.165	0.040 OR: 5.162 95% CI: 1.078–24.720	0.219 OR: 3.545 95% CI: 0.472–26.628
Uncontrolled	0 (0%)	2 (1.98%)	2 (3.90%)	0 (0%)	0.369 OR: 4.045 95% CI: 0.191–85.468	0.283 OR: 5.331 95% CI: 0.252–112.863	0.555 OR: 3.286 95% CI: 0.063–169.964
Any kind of autoimmune disease (SLE/APS/SS/RA/T1DM/other)	0 (0%)	13 (18.87%) 2 APS 1 SS 1 RA 1 T1DM 5 AIT 1 psoriasis 1 UCTD 1 Henoch-Schonlein purpura	11 (14.29%) 2 APS 1 RA 1 T1DM 4 AIT 1 psoriasis 1 UCTD 1 Henoch-Schonlein purpura	2 (8.33%) 1 SS 1 AIT	0.027 OR: 24.559 95% CI: 1.437–419.855	0.022 OR: 27.842 95% CI: 1.610–481.349	0.066 OR: 17.889 95% CI: 0.829–386.202
Thrombophilia gene mutations	0 (0%)	18 (17.82%)	13 (16.88%)	5 (20.83%)	0.013 OR: 35.671 95% CI: 2.114–601.862	0.015 OR: 33.698 95% CI: 1.965–577.721	0.011 OR: 45.410 95% CI: 2.408–856.463
Uterine fibroids or abnormal-shaped womb	0 (0%)	16 (15.84%) 12 uterine fibroids 4 abnormal-shaped womb	12 (15.58%) 8 uterine fibroids 4 abnormal-shaped womb	4 (16.67%) 4 uterine fibroids	0.017 OR: 31.070 95% CI: 1.834–526.478	0.018 OR: 30.725 95% CI: 1.785–528.788	0.018 OR: 35.341 95% CI: 1.828–683.218
Cervical incompetence	0 (0%)	4 (3.96%)	1 (1.30%)	3 (12.50%)	0.181 OR: 7.431 95% CI: 0.394–140.092	0.483 OR: 3.157 95% CI: 0.127–78.689	0.033 OR: 26.209 95% CI: 1.303–527.060
Polycystic ovary syndrome	1 (1.25%)	2 (1.98%)	2 (2.60%)	0 (0%)	0.705 OR: 1.596 95% CI: 0.142–17.924	0.546 OR: 2.107 95% CI: 0.187–23.720	0.962 OR: 1.082 95% CI: 0.043–27.413
Endometriosis	1 (1.25%)	3 (2.97%)	3 (3.90%)	0 (0%)	0.448 OR: 2.418 95% CI: 0.247–23.704	0.318 OR: 3.203 95% CI: 0.326–31.479	0.962 OR: 1.082 95% CI: 0.043–27.413
Smoking (currently or in the past)	3 (3.75%)	9 (8.91%)	5 (6.49%)	4 (16.67%)	0.178 OR: 2.511 95% CI: 0.657–9.601	0.440 OR: 1.782 95% CI: 0.411–7.729	0.042 OR: 5.133 95% CI: 1.062–24.816
Nulliparity	40 (50.0%)	55 (54.45%)	36 (46.75%)	19 (79.17%)	0.551 OR: 1.196 95% CI: 0.664–2.152	0.684 OR: 0.878 95% CI: 0.469–1.643	0.015 OR: 3.800 95% CI: 1.293–11.170
Previous pregnancy-related complications							
Previous GH or PE	0 (0%)	1 (0.99%)	1 (1.30%)	0 (0%)	0.593 OR: 2.403 95% CI: 0.096–59.786	0.483 OR: 3.157 95% CI: 0.127–78.689	0.555 OR: 3.286 95% CI: 0.063–169.964
Previous SGA or FGR	0 (0%)	1 (0.99%)	1 (1.30%)	0 (0%)	0.593 OR: 2.403 95% CI: 0.096–59.786	0.483 OR: 3.157 95% CI: 0.127–78.689	0.555 OR: 3.286 95% CI: 0.063–169.964
Previous GDM	1 (1.25%)	1 (0.99%)	0 (0%)	1 (4.17%)	0.868 OR: 0.790 95% CI: 0.049–12.830	0.513 OR: 0.342 95% CI: 0.014–8.523	0.389 OR: 3.435 95% CI: 0.207–57.079
Previous preterm delivery	0 (0%)	3 (2.97%)	2 (2.60%)	1 (4.17%)	0.251 OR: 5.721 95% CI: 0.291–112.387	0.283 OR: 5.331 95% CI: 0.253–112.863	0.158 OR: 10.277 95% CI: 0.405–260.697
History of miscarriage (spontaneous loss of a pregnancy before 20 weeks of gestation)	16 (20.0%)	31 (30.69%)	26 (33.77%)	5 (20.83%)	0.105 OR: 1.771 95% CI: 0.887–3.539	0.053 OR: 2.039 95% CI: 0.989–4.203	0.929 OR: 1.053 95% CI: 0.341–3.250
Recurrent pregnancy loss (2 or more consecutive losses of pregnancy occurring before 20 weeks)	4 (5.0%)	6 (5.94%)	2 (2.60%)	4 (16.67%)	0.783 OR: 1.200 95% CI: 0.327–4.406	0.440 OR: 0.507 95% CI: 0.090–2.850	0.075 OR: 3.800 95% CI: 0.873–16.541
History of perinatal death (the death of a baby between 20 weeks of gestation (or weighing more than 350 g) and 7 days after birth)	0 (0%)	6 (5.94%)	4 (5.19%)	2 (8.33%)	0.105 OR: 10.958 95% CI: 0.608–197.516	0.127 OR: 9.857 95% CI: 0.523–186.243	0.066 OR: 17.889 95% CI: 0.829–386.202
Method of conception							
Spontaneous	78 (97.5%)	90 (89.11%)	69 (89.61%)	21 (87.50%)	0.046 OR: 4.767 95% CI: 1.025–22.165	0.062 OR: 4.522 95% CI: 0.929–22.019	0.069 OR: 5.571 95% CI: 0.873–35.539
Assisted (IVF/ICSI/other)	2 (2.5%)	11 (10.89%)	8 (10.39%)	3 (12.50%)			
*Pregnancy details (first trimester of gestation)*							
Maternal age (years)	32 (25–42)	35 (21–57)	35 (21–57)	34 (21–42)	<0.001	<0.001	1.0
Advanced maternal age (≥35 years old at early stages of gestation)	18 (22.50%)	56 (55.45%)	46 (59.74%)	10 (41.67%)	<0.001 OR: 4.286 95% CI: 2.226–8.254	<0.001 OR: 5.111 95% CI: 2.551–10.240	0.068 OR: 2.460 95% CI: 0.936–6.467
BMI (kg/m^2^)	21.28 (17.16–29.76)	21.97 (16.71–52.47)	21.51 (17.47–52.47)	23.94 (16.71–33.13)	0.152	1.0	0.052
BMI ≥ 30 kg/m^2^	0 (0%)	10 (9.90%)	8 (10.39%)	2 (8.33%)	0.045 OR: 18.475 95% CI: 1.066–320.312	0.042 OR: 19.691 95% CI: 1.116–347.360	0.066 OR: 17.889 95% CI: 0.829–386.202
GA at time of blood sampling (weeks)	10.29 (9.57–13.71)	10.43 (9.29–13.71)	10.43 (9.29–13.71)	10.29 (9.71–13.71)	0.173	0.740	0.758
MAP (mmHg)	88.75 (67.67–103.83)	91.21 (72.92–110.58)	92.0 (72.92–110.58)	88.75 (73.92–105.50)	0.143	0.610	0.869
MAP (MoM)	1.05 (0.84–1.25)	1.09 (0.85–1.22)	1.09 (0.85–1.22)	1.07 (0.89–1.22)	0.282	0.856	1.0
Mean UtA-PI	1.39 (0.56–2.43)	1.58 (0.69–24.0)	1.60 (0.69–24.0)	1.56 (0.78–2.63)	0.054	0.394	0.357
Mean UtA-PI (MoM)	0.90 (0.37–1.55)	1.04 (0.42–1.71)	1.04 (0.42–1.67)	1.05 (0.52–1.71)	0.057	0.324	0.464
PIGF serum levels (pg/mL)	27.1 (8.1–137.0)	21.0 (3.60–57.40)	20.9 (3.60–57.40)	25.2 (5.20–45.40)	<0.001	<0.001	0.246
PIGF serum levels (MoM)	1.04 (0.38–2.61)	0.78 (0.17–1.58)	0.78 (0.25–1.56)	0.80 (0.17–1.58)	<0.001	0.004	0.017
PAPP-A serum levels (IU/L)	1.49 (0.48–15.69)	1.21 (0.07–9.21)	1.15 (0.07–9.21)	1.40 (0.33–5.03)	0.013	0.031	0.823
PAPP-A serum levels (MoM)	1.17 (0.37–3.18)	0.86 (0.27–3.08)	0.87 (0.27–3.08)	0.85 (0.32–2.49)	0.003	0.251	0.009
Free b-hCG serum levels (μg/L)	60.21 (9.9–200.6)	34.01 (5.40–239.1)	29.48 (5.40–177.1)	47.13 (10.01–239.1)	<0.001	<0.001	0.137
Free b-hCG serum levels (MoM)	1.02 (0.31–3.57)	0.86 (0.08–20.08)	0.67 (0.08–20.08)	0.95 (0.27–3.78)	0.013	0.053	0.409
Screen-positive for PE and/or FGR by FMF algorithm	0 (0%)	13 (12.87%)	6 (7.79%)	7 (29.17%)	0.027 OR: 24.559 95% CI: 1.437–419.855	0.069 OR: 14.636 95% CI: 0.810–264.432	0.004 OR: 69.000 95% CI: 3.762–1265.434
Screen-positive for preterm delivery (<34 weeks) by FMF algorithm	5 (6.25%)	16 (15.84%)	7 (9.09%)	9 (37.50%)	0.053 OR: 2.823 95% CI: 0.987–8.078	0.505 OR: 1.500 95% CI: 0.455–4.945	<0.001 OR: 9.000 95% CI: 2.642–30.661
Aspirin intake during pregnancy	0 (0%)	11 (10.89%)	6 (7.79%)	5 (20.83%)	0.038 OR: 20.459 95% CI: 1.186–352.748	0.069 OR: 14.636 95% CI: 0.810–264.432	0.011 OR: 45.410 95% CI: 2.408–856.463
*Pregnancy details*							
GA at miscarriage (weeks)	-	-	12.86 (9.71–19.86)	-	-	-	-
GA at stillbirth (weeks)	-	-	-	27.65 (20.71–38.71)	-	-	-
Invasive prenatal testing (CVS, AMC)	0 (0%)	5 (4.95%)	1 (1.30%)	4 (16.67%)	0.135 OR: 9.176 95% CI: 0.500–168.477	0.483 OR: 3.157 95% CI: 0.127–78.689	0.018 OR: 35.341 95% CI: 1.828–683.218

Continuous variables compared using Mann–Whitney or Kruskal–Wallis test are presented as medians (range). Categorical variables, presented as numbers (percent), are compared using odds ratio test. *p*-value ^1, 2, 3^: the comparison among normal term pregnancies, miscarriages, and stillbirths, and the comparison among normal term pregnancies and either miscarriages or stillbirths, respectively. T1DM, type 1 diabetes mellitus; T2DM, type 2 diabetes mellitus; WPW syndrome, Wolff-Parkinson-White syndrome; SLE, systemic lupus erythematosus; APS, antiphospholipid syndrome; SS, systemic scleroderma; RA, rheumatoid arthritis; AIT, autoimmune thyroid disease; UCTD, undifferentiated connective tissue disease; GH, gestational hypertension; PE, preeclampsia; FGR, fetal growth restriction; SGA, small for gestational age fetus; GDM, gestational diabetes mellitus; IVF, in vitro fertilization; ICSI, intracytoplasmic sperm injection; BMI, body mass index; MAP, mean arterial pressure; UtA-PI, uterine artery pulsatility index; PIGF, placental growth factor; PAPP-A, pregnancy-associated plasma protein-A; b-hCG–beta-subunit of human chorionic gonadotropin; FMF, Fetal Medicine Foundation; CVS, chorionic villus sampling; AMC, amniocentesis.

## Data Availability

The data presented in this study are available on request from the corresponding author. The data are not publicly available due to rights reserved by funding supporters.

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
