# Peer review of "First-Trimester Screening for Miscarriage or Stillbirth—Prediction Model Based on MicroRNA Biomarkers"

_ijms, 2023, doi:10.3390/ijms241210137_

Round 1
Reviewer 1 Report
Summary: The authors studied microRNAs associated with cardiovascular diseases by RT-PCR to predict the occurrence of miscarriage or stillbirth. As a result, screening models were obtained to predict miscarriage and/or stillbirth based on a combination of microRNAs.
Major comments:
1. There is a lack of comparison between the control group and groups with unexplained miscarriages and stillbirth. These groups are the most interesting from a clinical point of view.
2. Could the differences in age and BMI (p<0.05) affect the results of microRNA expression? Since the influence of age and BMI on the microRNA profile is known in other studies.
3. Table 1 was previously presented (https://www.mdpi.com/1422-0067/24/6/5177) and should be deleted.
4.
Minor comments:
1. In the abstract, the samples size of the studied groups by RT-PCR should be detailed.
2. The authors conducted a series of experiments to study the spectrum of microRNAs in early studies. It would be good to propose an algorithm for early microRNA screening to predict major obstetric complications.
Reviewer 2 Report
I read this with interest as a clinician. I would have lighted a slightly more detailed explanation of the mRNA. I appreciate that appropriate references were put in the article but a one or two sentence explanation would have made my reading easier.For instance I do not really understand if the techniques state presence or absence of mRNA or is there a quantitative element to the result?
Likewise with the methods could have been clearer . Again I acknowledge that you have stated that these are described in another publication which I have not accessed. Were these samples taken at "booking" at say 12 weeks gestation or was there a variation of gestational timing from those that miscarried and those that did not. Those that conceived with IVF are likely to have known earlier than those that conceived spontaneously that they were pregnant.
The implications of these data should have been discussed further. I do not see any evidence of benefit into routinely testing for mRNAs to predict miscarriage. There may be some benefit to help inform a decision to deliver early by induction or caesarean section to prevent stillbirth. That would require prospective studies.
The other research that may be interesting is the measurement sequentially throughout pregnancy of different mRNA presence.
So in summary this is a very interesting paper that would benefit of a slightly stronger explanation of the background for clinicians who are less strong than maybe you anticipate with the science. The implications on clinical care and future research should be discussed.
